# Bipolar Fuzzy UP-Algebras

**Korawut Kawila, Chaiphon Udomsetchai and Aiyared Iampan \*** 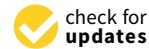

Department of Mathematics, School of Science, University of Phayao, Phayao 56000, Thailand;
korawutkawila@gmail.com (K.K.); lovesumen3@gmail.com (C.U.)

\* Correspondence: aiyared.ia@up.ac.th; Tel.: +66-054-466-666

**Abstract:** In this paper, we apply the notion of bipolar-valued fuzzy set to UP-algebras. We introduce the notions of bipolar fuzzy UP-subalgebras (resp., bipolar fuzzy UP-filters, bipolar fuzzy UP-ideals, and bipolar fuzzy strongly UP-ideals) of UP-algebras and prove their generalizations. We provide a condition for a bipolar fuzzy UP-filter to be a bipolar fuzzy UP-ideal. Further, we discuss the relation between bipolar fuzzy UP-subalgebras (resp., bipolar fuzzy UP-filters, bipolar fuzzy UP-ideals, and bipolar fuzzy strongly UP-ideals) and their level cuts.

**Keywords:** UP-algebra; bipolar fuzzy UP-subalgebra; bipolar fuzzy UP-filter; bipolar fuzzy UP-ideal; bipolar fuzzy strongly UP-ideal

**MSC:** 03G25; 08A72

## 1. Introduction

Among the important and intensively studied classes of algebras are the algebras of logic. Examples of these are BCK-algebras [1], BCI-algebras [2], BCH-algebras [3], KU-algebras [4], SU-algebras [5], UP-algebras [6], and others. They are strongly connected with logic. For example, the BCI-algebras introduced by Iséki [2] in 1966 have connections with BCI-logic, being the BCI-system in combinatory logic, which has applications in the language of functional programming. BCK- and BCI-algebras are two classes of logical algebras. They were introduced by Imai and Iséki [1,2] in 1966 and have been extensively investigated by many researchers. It is known that the class of BCK-algebras is a proper subclass of the class of BCI-algebras.

The notion of UP-algebras was introduced by Iampan [6] in 2017, and it is known that the class of KU-algebras [4] is a proper subclass of the class of UP-algebras. They have been examined by several researchers, for example, Somjanta et al. [7] introduced the notion of fuzzy sets in UP-algebras, the notion of intuitionistic fuzzy sets in UP-algebras was introduced by Kesorn et al. [8], the notion of *Q*-fuzzy sets in UP-algebras was introduced by Tanamoon et al. [9], Senapati et al. [10,11] applied cubic set and interval-valued intuitionistic fuzzy structure to UP-algebras, and so on.

The notion of fuzzy subsets of a set was first considered by Zadeh [12] in 1965. The fuzzy set theories developed by Zadeh and others have found many applications in the domain of mathematics and elsewhere. There are several kinds of fuzzy set extensions in the fuzzy set theory, for example, intuitionistic fuzzy sets (see [8,13,14]), interval-valued fuzzy sets (see [15]), vague sets (see [16]), bipolar-valued fuzzy sets, etc. The notion of bipolar-valued fuzzy sets, first introduced by Zhang [17] in 1994, is an extension of fuzzy sets whose membership degree range is enlarged from the interval $[0, 1]$ to $[-1, 0]$.

After the introduction of the notion of bipolar-valued fuzzy sets by Zhang [17], several studies were conducted on the generalizations of the notion of bipolar-valued fuzzy sets and the application to many logical algebras, a few of which are mentioned in the following. Jun et al. (see [18,19]) introduced

the notions of bipolar fuzzy subalgebras, bipolar fuzzy closed ideals, bipolar fuzzy regularities, bipolar fuzzy regular subalgebras, bipolar fuzzy filters, and bipolar fuzzy closed quasi filters in BCH-algebras. Akram et al. studied bipolar-valued fuzzy set theory in graphs, Lie algebras, and K-algebras (see [20–22]). Muhiuddin [23] introduced the notions of bipolar fuzzy KU-subalgebras and bipolar fuzzy KU-ideals in KU-algebras. Senapati introduced the notion of bipolar fuzzy BG-subalgebras in BG-algebras and the notion of bipolar fuzzy B-subalgebras in B-algebras (see [24,25]).

In this paper, we apply the notion of bipolar-valued fuzzy set to UP-algebras. We introduce the notions of bipolar fuzzy UP-subalgebras (resp., bipolar fuzzy UP-filters, bipolar fuzzy UP-ideals, and bipolar fuzzy strongly UP-ideals) of UP-algebras and prove their generalizations. We provide a condition for a bipolar fuzzy UP-filter to be a bipolar fuzzy UP-ideal. Further, we discuss the relation between bipolar fuzzy UP-subalgebras (resp., bipolar fuzzy UP-filters, bipolar fuzzy UP-ideals, and bipolar fuzzy strongly UP-ideals) and their level cuts.

## 2. Basic Results on UP-Algebras

An algebra $A = (A, \cdot, 0)$ of type $(2, 0)$ is called a *UP-algebra* [6], where $A$ is a nonempty set, $\cdot$ is a binary operation on $A$, and 0 is a fixed element of $A$ (i.e., a nullary operation) if it satisfies the following axioms: for any $x, y, z \in A$,

**(UP-1)** $(y \cdot z) \cdot ((x \cdot y) \cdot (x \cdot z)) = 0$,
**(UP-2)** $0 \cdot x = x$,
**(UP-3)** $x \cdot 0 = 0$, and
**(UP-4)** $x \cdot y = 0$ and $y \cdot x = 0$ imply $x = y$.

From [6], we know that the notion of UP-algebras is a generalization of KU-algebras.

**Example 1** ([6]). *Let X be a universal set. Define two binary operations $\cdot$ and $*$ on the power set of X by putting $A \cdot B = B \cap A'$ and $A * B = B \cup A'$ for all $A, B \in \mathcal{P}(X)$. Then, $(\mathcal{P}(X), \cdot, \varnothing)$ and $(\mathcal{P}(X), *, X)$ are UP-algebras, and we shall call it the power UP-algebra of type 1 and the power UP-algebra of type 2, respectively.*

In a UP-algebra $A = (A, \cdot, 0)$, the following assertions are valid (see [6,26]).

$$(\forall x \in A)(x \cdot x = 0), \tag{1}$$

$$(\forall x, y, z \in A)(x \cdot y = 0, y \cdot z = 0 \Rightarrow x \cdot z = 0), \tag{2}$$

$$(\forall x, y, z \in A)(x \cdot y = 0 \Rightarrow (z \cdot x) \cdot (z \cdot y) = 0), \tag{3}$$

$$(\forall x, y, z \in A)(x \cdot y = 0 \Rightarrow (y \cdot z) \cdot (x \cdot z) = 0), \tag{4}$$

$$(\forall x, y \in A)(x \cdot (y \cdot x) = 0), \tag{5}$$

$$(\forall x, y \in A)((y \cdot x) \cdot x = 0 \Leftrightarrow x = y \cdot x), \tag{6}$$

$$(\forall x, y \in A)(x \cdot (y \cdot y) = 0), \tag{7}$$

$$(\forall a, x, y, z \in A)((x \cdot (y \cdot z)) \cdot (x \cdot ((a \cdot y) \cdot (a \cdot z))) = 0), \tag{8}$$

$$(\forall a, x, y, z \in A)((((a \cdot x) \cdot (a \cdot y)) \cdot z) \cdot ((x \cdot y) \cdot z) = 0), \tag{9}$$

$$(\forall x, y, z \in A)(((x \cdot y) \cdot z) \cdot (y \cdot z) = 0), \tag{10}$$

$$(\forall x, y, z \in A)(x \cdot y = 0 \Rightarrow x \cdot (z \cdot y) = 0), \tag{11}$$

$$(\forall x, y, z \in A)(((x \cdot y) \cdot z) \cdot (x \cdot (y \cdot z)) = 0), \text{ and} \tag{12}$$

$$(\forall a, x, y, z \in A)(((x \cdot y) \cdot z) \cdot (y \cdot (a \cdot z)) = 0). \tag{13}$$

**Definition 1** ([6,7]). *A nonempty subset S of a UP-algebra $(A, \cdot, 0)$ is called*

*(1)     a UP-subalgebra of A if for any $x, y \in S$, $x \cdot y \in S$.*
*(2)     a UP-filter of A if it satisfies the following properties:*

(i) the constant 0 of A is in S, and

(ii) for any $x, y \in A$, $x \cdot y \in S$ and $x \in S$ imply $y \in S$.

(3) a UP-ideal of A if it satisfies the following properties:

(i) the constant 0 of A is in S, and

(ii) for any $x, y, z \in A$, $x \cdot (y \cdot z) \in S$ and $y \in S$ imply $x \cdot z \in S$.

(4) a strongly UP-ideal of A if it satisfies the following properties:

(i) the constant 0 of A is in S, and

(ii) for any $x, y, z \in A$, $(z \cdot y) \cdot (z \cdot x) \in S$ and $y \in S$ imply $x \in S$.

Guntasow et al. [27] proved the generalization that the notion of UP-subalgebras is a generalization of UP-filters, the notion of UP-filters is a generalization of UP-ideals, and the notion of UP-ideals is a generalization of strongly UP-ideals. Moreover, they also proved that a UP-algebra $A$ is the only one strongly UP-ideal of itself.

In what follows, let $A$ be a UP-algebra unless otherwise specified.

The following theorem is easily obtained.

**Theorem 1.** *Let $\{B_i\}_{i \in I}$ be a nonempty family of UP-subalgebras (resp., UP-filters, UP-ideals, strongly UP-ideals) of A. Then, $\cap_{i \in I} B_i$ is a UP-subalgebra (resp., UP-filters, UP-ideals, strongly UP-ideals) of A.*

## 3. Bipolar Fuzzy Sets

Let $X$ be the universe of discourse. A *bipolar-valued fuzzy set* [17] $\varphi$ in $X$ is an object having the form

$$\varphi = \{(x, \varphi^-(x), \varphi^+(x)) \mid x \in X\}$$

where $\varphi^- : X \to [-1, 0]$ and $\varphi^+ : X \to [0, 1]$ are mappings. For the sake of simplicity, we shall use the symbol $\varphi = (X; \varphi^-, \varphi^+)$ for the bipolar-valued fuzzy set $\varphi = \{(x, \varphi^-(x), \varphi^+(x)) \mid x \in X\}$, and use the notion of bipolar fuzzy sets instead of the notion of bipolar-valued fuzzy sets.

Next, we introduce the notion of bipolar fuzzy UP-subalgebras (resp., bipolar fuzzy UP-filters, bipolar fuzzy UP-ideals, and bipolar fuzzy strongly UP-ideals) of a UP-algebra $A$ and provide the necessary examples.

**Definition 2.** *A bipolar fuzzy set $\varphi = (A; \varphi^-, \varphi^+)$ in A is called a bipolar fuzzy UP-subalgebra of A if it satisfies the following properties: for any $x, y \in A$,*

(1) $\varphi^-(x \cdot y) \leq \max\{\varphi^-(x), \varphi^-(y)\}$, *and*

(2) $\varphi^+(x \cdot y) \geq \min\{\varphi^+(x), \varphi^+(y)\}$.

**Remark 1.** *If $\varphi = (A; \varphi^-, \varphi^+)$ is a bipolar fuzzy UP-subalgebra of A, then,*

$$\varphi^-(0) \leq \varphi^-(x) \text{ and } \varphi^+(0) \geq \varphi^+(x) \text{ for all } x \in A.$$

*Indeed, for all $x \in A$,*

$$\varphi^-(0) = \varphi^-(x \cdot x) \leq \max\{\varphi^-(x), \varphi^-(x)\} = \varphi^-(x)$$

*and*

$$\varphi^+(0) = \varphi^+(x \cdot x) \geq \min\{\varphi^+(x), \varphi^+(x)\} = \varphi^+(x).$$

**Example 2.** *Consider a UP-algebra $A = \{0, 1, 2, 3\}$ with the following Cayley table:*

| · | 0 | 1 | 2 | 3 |
|---|---|---|---|---|
| 0 | 0 | 1 | 2 | 3 |
| 1 | 0 | 0 | 0 | 3 |
| 2 | 0 | 1 | 0 | 3 |
| 3 | 0 | 1 | 0 | 0 |

*Define a bipolar fuzzy set $\varphi = (A; \varphi^-, \varphi^+)$ in A as follows:*

| A | 0 | 1 | 2 | 3 |
|---|---|---|---|---|
| $\varphi^-$ | −0.8 | −0.6 | −0.2 | −0.1 |
| $\varphi^+$ | 0.9 | 0.7 | 0.5 | 0.4 |

*Then, $\varphi = (A; \varphi^-, \varphi^+)$ is a bipolar fuzzy UP-subalgebra of A.*

**Definition 3.** *A bipolar fuzzy set $\varphi = (A; \varphi^-, \varphi^+)$ in A is called a bipolar fuzzy UP-filter of A if it satisfies the following properties: for any $x, y \in A$,*

(1)   $\varphi^-(0) \leq \varphi^-(x)$,
(2)   $\varphi^+(0) \geq \varphi^+(x)$,
(3)   $\varphi^-(y) \leq \max\{\varphi^-(x \cdot y), \varphi^-(x)\}$, *and*
(4)   $\varphi^+(y) \geq \min\{\varphi^+(x \cdot y), \varphi^+(x)\}$.

**Example 3.** *Consider a UP-algebra $A = \{0, 1, 2, 3\}$ with the following Cayley table:*

| · | 0 | 1 | 2 | 3 |
|---|---|---|---|---|
| 0 | 0 | 1 | 2 | 3 |
| 1 | 0 | 0 | 2 | 3 |
| 2 | 0 | 1 | 0 | 3 |
| 3 | 0 | 1 | 2 | 0 |

*Define a bipolar fuzzy set $\varphi = (A; \varphi^-, \varphi^+)$ in A as follows:*

| A | 0 | 1 | 2 | 3 |
|---|---|---|---|---|
| $\varphi^-$ | −0.7 | −0.3 | −0.4 | −0.6 |
| $\varphi^+$ | 0.9 | 0.5 | 0.1 | 0.2 |

*Then, $\varphi = (A; \varphi^-, \varphi^+)$ is a bipolar fuzzy UP-filter of A.*

**Definition 4.** *A bipolar fuzzy set $\varphi = (A; \varphi^-, \varphi^+)$ in A is called a bipolar fuzzy UP-ideal of A if it satisfies the following properties: for any $x, y, z \in A$,*

(1)   $\varphi^-(0) \leq \varphi^-(x)$,
(2)   $\varphi^+(0) \geq \varphi^+(x)$,
(3)   $\varphi^-(x \cdot z) \leq \max\{\varphi^-(x \cdot (y \cdot z)), \varphi^-(y)\}$, *and*
(4)   $\varphi^+(x \cdot z) \geq \min\{\varphi^+(x \cdot (y \cdot z)), \varphi^+(y)\}$.

**Example 4.** *Consider a UP-algebra $A = \{0, 1, 2, 3, 4\}$ with the following Cayley table:*

$$
\begin{array}{c|ccccc}
\cdot & 0 & 1 & 2 & 3 & 4 \\
\hline
0 & 0 & 1 & 2 & 3 & 4 \\
1 & 0 & 0 & 2 & 3 & 4 \\
2 & 0 & 0 & 0 & 3 & 4 \\
3 & 0 & 0 & 0 & 0 & 4 \\
4 & 0 & 0 & 0 & 1 & 0 \\
\end{array}
$$

*Define a bipolar fuzzy set $\varphi = (A; \varphi^-, \varphi^+)$ in A as follows:*

| $A$ | 0 | 1 | 2 | 3 | 4 |
|---|---|---|---|---|---|
| $\varphi^-$ | $-0.8$ | $-0.5$ | $-0.5$ | $-0.2$ | $-0.2$ |
| $\varphi^+$ | 0.9 | 0.6 | 0.6 | 0.4 | 0.4 |

*Then, $\varphi = (A; \varphi^-, \varphi^+)$ is a bipolar fuzzy UP-ideal of A.*

**Definition 5.** *A bipolar fuzzy set $\varphi = (A; \varphi^-, \varphi^+)$ in A is called a bipolar fuzzy strongly UP-ideal of A if it satisfies the following properties: for any $x, y, z \in A$,*

(1)    $\varphi^-(0) \le \varphi^-(x)$,
(2)    $\varphi^+(0) \ge \varphi^+(x)$,
(3)    $\varphi^-(x) \le \max\{\varphi^-((z \cdot y) \cdot (z \cdot x)), \varphi^-(y)\}$, *and*
(4)    $\varphi^+(x) \ge \min\{\varphi^+((z \cdot y) \cdot (z \cdot x)), \varphi^+(y)\}$.

**Example 5.** *Consider a UP-algebra $A = \{0, 1, 2, 3\}$ with the following Cayley table:*

$$
\begin{array}{c|cccc}
\cdot & 0 & 1 & 2 & 3 \\
\hline
0 & 0 & 1 & 2 & 3 \\
1 & 0 & 0 & 2 & 2 \\
2 & 0 & 1 & 0 & 3 \\
3 & 0 & 1 & 0 & 0 \\
\end{array}
$$

*Define a bipolar fuzzy set $\varphi = (A; \varphi^-, \varphi^+)$ in A as follows:*

| $A$ | 0 | 1 | 2 | 3 |
|---|---|---|---|---|
| $\varphi^-$ | $-0.5$ | $-0.5$ | $-0.5$ | $-0.5$ |
| $\varphi^+$ | 0.8 | 0.8 | 0.8 | 0.8 |

*Then, $\varphi = (A; \varphi^-, \varphi^+)$ is a bipolar fuzzy strongly UP-ideal of A.*

**Theorem 2.** *A bipolar fuzzy set $\varphi = (A; \varphi^-, \varphi^+)$ in A is constant if and only if it is a bipolar fuzzy strongly UP-ideal of A.*

**Proof.** Assume that $\varphi = (A; \varphi^-, \varphi^+)$ is a constant bipolar fuzzy set in $A$. Then, there exist $l \in [-1, 0]$ and $k \in [0, 1]$ such that

$$\varphi^-(x) = l \text{ for all } x \in A \text{ and } \varphi^+(x) = k \text{ for all } x \in A.$$

Thus, $\varphi^-(0) = l \le l = \varphi^-(x)$ and $\varphi^+(0) = k \ge k = \varphi^+(x)$ for all $x \in A$. For all $x, y, z \in A$,

$$\varphi^-(x) = l \le l = \max\{l, l\} = \max\{\varphi^-((z \cdot y) \cdot (z \cdot x)), \varphi^-(y)\}$$

and

$$\varphi^+(x) = k \ge k = \min\{k, k\} = \min\{\varphi^+((z \cdot y) \cdot (z \cdot x)), \varphi^+(y)\}.$$

Hence, $\varphi = (A; \varphi^-, \varphi^+)$ is a bipolar fuzzy strongly UP-ideal of $A$.

Conversely, assume that $\varphi = (A; \varphi^-, \varphi^+)$ is a bipolar fuzzy strongly UP-ideal of $A$. Then, for all $x, y, z \in A$,

$$\varphi^-(0) \leq \varphi^-(x) \text{ and } \varphi^+(0) \geq \varphi^+(x),$$

and

$$\varphi^-(x) \leq \max\{\varphi^-((z \cdot y) \cdot (z \cdot x)), \varphi^-(y)\} \text{ and } \varphi^+(x) \geq \min\{\varphi^+((z \cdot y) \cdot (z \cdot x)), \varphi^+(y)\}.$$

For all $x \in A$,

$$
\begin{aligned}
\varphi^-(x) &\leq \max\{\varphi^-((x \cdot 0) \cdot (x \cdot x)), \varphi^-(0)\} \\
&\leq \max\{\varphi^-(0 \cdot 0), \varphi^-(0)\} && \text{((UP-3), (1))} \\
&\leq \max\{\varphi^-(0), \varphi^-(0)\} && \text{((UP-2))} \\
&= \varphi^-(0).
\end{aligned}
$$

and

$$
\begin{aligned}
\varphi^+(x) &\geq \min\{\varphi^+((x \cdot 0) \cdot (x \cdot x)), \varphi^+(0)\} \\
&= \min\{\varphi^+(0 \cdot 0), \varphi^+(0)\} && \text{((UP-3), (1))} \\
&= \min\{\varphi^+(0), \varphi^+(0)\} && \text{((UP-2))} \\
&= \varphi^+(0).
\end{aligned}
$$

Hence, $\varphi^-(x) = \varphi^-(0)$ and $\varphi^+(x) = \varphi^+(0)$ for all $x \in A$. Therefore, $\varphi = (A; \varphi^-, \varphi^+)$ is a constant bipolar fuzzy set in $A$. $\square$

**Theorem 3.** *Every bipolar fuzzy strongly UP-ideal of $A$ is a bipolar fuzzy UP-ideal.*

**Proof.** Let $\varphi = (A; \varphi^-, \varphi^+)$ be a bipolar fuzzy strongly UP-ideal of $A$. By Theorem 2, there exists $(l, k) \in [-1, 0] \times [0, 1]$ such that

$$\varphi^-(x) = l \text{ and } \varphi^+(x) = k \text{ for all } x \in A.$$

For all $x, y, z \in A$,

$$\varphi^-(0) = l \leq l = \varphi^-(x)$$

and

$$\varphi^-(x \cdot z) = l \leq l = \max\{l, l\} = \max\{\varphi^-(x \cdot (y \cdot z)), \varphi^-(y)\},$$

and

$$\varphi^+(0) = k \geq k = \varphi^+(x)$$

and

$$\varphi^+(x \cdot z) = k \geq k = \min\{k, k\} = \min\{\varphi^+(x \cdot (y \cdot z)), \varphi^+(y)\}.$$

Hence, $\varphi = (A; \varphi^-, \varphi^+)$ is a bipolar fuzzy UP-ideal of $A$. $\square$

**Example 6.** *Consider a UP-algebra $A = \{0, 1, 2, 3, 4\}$ with the following Cayley table:*

| · | 0 | 1 | 2 | 3 | 4 |
|---|---|---|---|---|---|
| 0 | 0 | 1 | 2 | 3 | 4 |
| 1 | 0 | 0 | 2 | 3 | 4 |
| 2 | 0 | 0 | 0 | 3 | 4 |
| 3 | 0 | 1 | 1 | 0 | 4 |
| 4 | 0 | 1 | 2 | 3 | 0 |

*Define a bipolar fuzzy set $\varphi = (A; \varphi^-, \varphi^+)$ in A as follows:*

| $A$ | 0 | 1 | 2 | 3 | 4 |
|---|---|---|---|---|---|
| $\varphi^-$ | $-0.9$ | $-0.6$ | $-0.5$ | $-0.2$ | $-0.7$ |
| $\varphi^+$ | 0.8 | 0.5 | 0.2 | 0.1 | 0.5 |

*Then, $\varphi = (A; \varphi^-, \varphi^+)$ is a bipolar fuzzy UP-ideal of A but it is not a bipolar fuzzy strongly UP-ideal of A. Indeed,*

$$\varphi^-(3) = -0.2 > -0.7 = \max\{\varphi^-((3 \cdot 4) \cdot (3 \cdot 3)), \varphi^-(4)\}$$

*and*

$$\varphi^+(3) = 0.1 < 0.5 = \min\{\varphi^+((3 \cdot 4) \cdot (3 \cdot 3)), \varphi^+(4)\}.$$

**Theorem 4.** *Every bipolar fuzzy UP-ideal of A is a bipolar fuzzy UP-filter.*

**Proof.** Let $\varphi$ be a bipolar fuzzy UP-ideal of A. Then, for all $x, y \in A$, $\varphi^-(0) \le \varphi^-(x)$ and

$$\varphi^-(y) = \varphi^-(0 \cdot y) \tag{(UP-2)}$$
$$\le \max\{\varphi^-(0 \cdot (x \cdot y)), \varphi^-(x)\}$$
$$= \max\{\varphi^-(x \cdot y), \varphi^-(x)\}, \tag{(UP-2)}$$

and $\varphi^+(0) \ge \varphi^+(x)$ and

$$\varphi^+(y) = \varphi^+(0 \cdot y) \tag{(UP-2)}$$
$$\ge \min\{\varphi^+(0 \cdot (x \cdot y)), \varphi^+(x)\}$$
$$= \min\{\varphi^+(x \cdot y), \varphi^+(x)\}. \tag{(UP-2)}$$

Hence, $\varphi$ is a bipolar fuzzy UP-filter of A.  □

**Example 7.** *Consider a UP-algebra $A = \{0, 1, 2, 3\}$ with the following Cayley table:*

| · | 0 | 1 | 2 | 3 |
|---|---|---|---|---|
| 0 | 0 | 1 | 2 | 3 |
| 1 | 0 | 0 | 2 | 2 |
| 2 | 0 | 1 | 0 | 3 |
| 3 | 0 | 0 | 0 | 0 |

*Define a bipolar fuzzy set $\varphi = (A; \varphi^-, \varphi^+)$ in A as follows:*

| $A$ | 0 | 1 | 2 | 3 |
|---|---|---|---|---|
| $\varphi^-$ | $-0.7$ | $-0.3$ | $-0.1$ | $-0.1$ |
| $\varphi^+$ | 0.8 | 0.5 | 0.2 | 0.2 |

*Then, $\varphi = (A; \varphi^-, \varphi^+)$ is a bipolar fuzzy UP-filter of A but it is not a bipolar fuzzy UP-ideal of A. Indeed,*

$$\varphi^-(2 \cdot 3) = \varphi^-(3) = -0.1 > -0.3 = \max\{\varphi^-(2 \cdot (1 \cdot 3)), \varphi^-(1)\}$$

*and*

$$\varphi^+(2 \cdot 3) = \varphi^+(3) = 0.2 < 0.5 = \min\{\varphi^+(2 \cdot (1 \cdot 3)), \varphi^+(1)\}.$$

**Theorem 5.** *Every bipolar fuzzy UP-filter of A is a bipolar fuzzy UP-subalgebra.*

**Proof.** Let $\varphi$ be a bipolar fuzzy UP-filter of $A$. Then, for all $x, y \in A$, $\varphi^-(0) \leq \varphi^-(x)$ and

$$\begin{aligned}
\varphi^-(x \cdot y) &\leq \max\{\varphi^-(y \cdot (x \cdot y)), \varphi^-(y)\} \\
&= \max\{\varphi^-(0), \varphi^-(y)\} \\
&\leq \max\{\varphi^-(x), \varphi^-(y)\},
\end{aligned} \tag{(5)}$$

and $\varphi^+(0) \geq \varphi^+(x)$ and

$$\begin{aligned}
\varphi^+(x \cdot y) &\geq \min\{\varphi^+(y \cdot (x \cdot y)), \varphi^+(y)\} \\
&= \min\{\varphi^+(0), \varphi^+(y)\} \\
&\geq \min\{\varphi^+(x), \varphi^+(y)\}.
\end{aligned} \tag{(5)}$$

Hence, $\varphi$ is a bipolar fuzzy UP-subalgebra of $A$.  $\square$

**Example 8.** *Consider a UP-algebra $A = \{0, 1, 2, 3\}$ with the following Cayley table:*

| $\cdot$ | 0 | 1 | 2 | 3 |
|---|---|---|---|---|
| 0 | 0 | 1 | 2 | 3 |
| 1 | 0 | 0 | 1 | 3 |
| 2 | 0 | 0 | 0 | 3 |
| 3 | 0 | 0 | 0 | 0 |

*Define a bipolar fuzzy set $\varphi = (A; \varphi^-, \varphi^+)$ in A as follows:*

| $A$ | 0 | 1 | 2 | 3 |
|---|---|---|---|---|
| $\varphi^+$ | 0.8 | 0.4 | 0.2 | 0.1 |
| $\varphi^-$ | $-0.9$ | $-0.5$ | $-0.3$ | $-0.2$ |

*Then, $\varphi = (A; \varphi^-, \varphi^+)$ is a bipolar fuzzy UP-subalgebra of A but it is not a bipolar fuzzy UP-filter of A. Indeed,*

$$\varphi^-(2) = -0.3 > -0.5 = \max\{\varphi^-(1 \cdot 2), \varphi^-(1)\}$$

*and*

$$\varphi^+(2) = 0.2 < 0.4 = \min\{\varphi^+(1 \cdot 2), \varphi^+(1)\}.$$

By Theorems 3–5 and Examples 6–8, we have that the notion of bipolar fuzzy UP-subalgebras is a generalization of bipolar fuzzy UP-filters, the notion of bipolar fuzzy UP-filters is a generalization of bipolar fuzzy UP-ideals, and the notion of bipolar fuzzy UP-ideals is a generalization of bipolar fuzzy strongly UP-ideals.

## 4. Level Cuts of a Bipolar Fuzzy Set

In this section, we discuss the relation between bipolar fuzzy UP-subalgebras (resp., bipolar fuzzy UP-filters, bipolar fuzzy UP-ideals, and bipolar fuzzy strongly UP-ideals) and their level cuts.

**Definition 6.** *Let $\varphi = (A; \varphi^-, \varphi^+)$ be a bipolar fuzzy set in A. For $(t^-, t^+) \in [-1, 0] \times [0, 1]$, the sets*

$$N_L(\varphi; t^-) = \{x \in A \mid \varphi^-(x) \leq t^-\}$$

*and*

$$P_U(\varphi; t^+) = \{x \in A \mid \varphi^+(x) \geq t^+\}$$

*are called the negative lower $t^-$-cut and the positive upper $t^+$-cut of $\varphi = (A; \varphi^-, \varphi^+)$, respectively. The set*

$$C(\varphi; (t^-, t^+)) = N_L(\varphi; t^-) \cap P_U(\varphi; t^+)$$

*is called the $(t^-, t^+)$-cut of $\varphi = (A; \varphi^-, \varphi^+)$. For any $k \in [0, 1]$, we denote the set*

$$C(\varphi; k) = C(\varphi; (-k, k)) = N_L(\varphi; -k) \cap P_U(\varphi; k)$$

*is called the k-cut of $\varphi = (A; \varphi^-, \varphi^+)$.*

**Example 9.** *From Example 8, we have $N_L(\varphi; -0.5) = \{0, 1\}, P_U(\varphi; 0.2) = \{0, 1, 2\}$, and $C(\varphi; (-0.5, 0.2)) = \{0, 1\}$.*

**Theorem 6.** *Let $\varphi = (A; \varphi^-, \varphi^+)$ be a bipolar fuzzy set in A. Then, $\varphi = (A; \varphi^-, \varphi^+)$ is a bipolar fuzzy UP-subalgebra of A if and only if the following statements are valid:*

(1)  *for all $t^- \in [-1, 0], N_L(\varphi; t^-)$ is a UP-subalgebra of A if $N_L(\varphi; t^-)$ is nonempty, and*
(2)  *for all $t^+ \in [0, 1], P_U(\varphi; t^+)$ is a UP-subalgebra of A if $P_U(\varphi; t^+)$ is nonempty.*

**Proof.** Assume that $\varphi$ is a bipolar fuzzy UP-subalgebra of $A$. Let $t^- \in [-1, 0]$ be such that $N_L(\varphi; t^-) \neq \emptyset$ and let $x, y \in N_L(\varphi; t^-)$. Then, $\varphi^-(x) \leq t^-$ and $\varphi^-(y) \leq t^-$. Since $\varphi$ is a bipolar fuzzy UP-subalgebra of $A$, we have $\varphi^-(x \cdot y) \leq \max\{\varphi^-(x), \varphi^-(y)\} \leq t^-$. Thus, $x \cdot y \in N_L(\varphi; t^-)$. Hence, $N_L(\varphi; t^-)$ is a UP-subalgebra of $A$. Next, let $t^+ \in [0, 1]$ be such that $P_U(\varphi; t^+) \neq \emptyset$ and let $x, y \in P_U(\varphi; t^+)$. Then, $\varphi^+(x) \geq t^+$ and $\varphi^+(y) \geq t^+$. Since $\varphi$ is a bipolar fuzzy UP-subalgebra of $A$, we have $\varphi^+(x \cdot y) \geq \min\{\varphi^+(x), \varphi^+(y)\} \geq t^+$. Thus, $x \cdot y \in P_U(\varphi; t^+)$. Hence, $P_U(\varphi; t^+)$ is a UP-subalgebra of $A$.

Conversely, assume that for all $t^- \in [0, 1], N_L(\varphi; t^-)$ is a UP-subalgebra of $A$ if $N_L(\varphi; t^-)$ is nonempty and for all $t^+ \in [0, 1], P_U(\varphi; t^+)$ is a UP-subalgebra of $A$ if $P_U(\varphi; t^+)$ is nonempty. Let $x, y \in A$. Then, $\varphi^-(x), \varphi^-(y) \in [-1, 0]$. Choose $t^- = \max\{\varphi^-(x), \varphi^-(y)\}$. Then, $\varphi^-(x) \leq t^-$ and $\varphi^-(y) \leq t^-$, that is, $x, y \in N_L(\varphi; t^-) \neq \emptyset$. By assumption, we have $N_L(\varphi; t^-)$ is a UP-subalgebra of $A$. So, $x \cdot y \in N_L(\varphi; t^-)$. Hence, $\varphi^-(x \cdot y) \leq t^- = \max\{\varphi^-(x), \varphi^-(y)\}$. Next, let $x, y \in A$. Then, $\varphi^+(x), \varphi^+(y) \in [0, 1]$. Choose $t^+ = \min\{\varphi^+(x), \varphi^+(y)\}$. Then, $\varphi^+(x) \geq t^+$ and $\varphi^+(y) \geq t^+$, that is, $x, y \in P_U(\varphi; t^+) \neq \emptyset$. By assumption, we have $P_U(\varphi; t^+)$ is a UP-subalgebra of $A$. So, $x \cdot y \in P_U(\varphi; t^+)$. Hence, $\varphi^+(x \cdot y) \geq t^+ = \min\{\varphi^+(x), \varphi^+(y)\}$. Therefore, $\varphi = (A; \varphi^-, \varphi^+)$ is a bipolar fuzzy UP-subalgebra of $A$. $\square$

**Corollary 1.** *If $\varphi = (A; \varphi^-, \varphi^+)$ is a bipolar fuzzy UP-subalgebra of A, then, for all $k \in [0, 1], C(\varphi; k)$ is a UP-subalgebra of A while $C(\varphi; k)$ is nonempty.*

**Proof.** It is straightforward by Theorems 1 and 6. $\square$

**Theorem 7.** *Let $\varphi = (A; \varphi^-, \varphi^+)$ be a bipolar fuzzy set in A. Then, $\varphi = (A; \varphi^-, \varphi^+)$ is a bipolar fuzzy UP-filter of A if and only if the following statements are valid:*

(1)  *for all $t^- \in [-1, 0], N_L(\varphi; t^-)$ is a UP-filter of A if $N_L(\varphi; t^-)$ is nonempty, and*
(2)  *for all $t^+ \in [0, 1], P_U(\varphi; t^+)$ is a UP-filter of A if $P_U(\varphi; t^+)$ is nonempty.*

**Proof.** Assume that $\varphi$ is a bipolar fuzzy UP-filter of $A$. Let $t^- \in [-1, 0]$ be such that $N_L(\varphi; t^-) \neq \emptyset$ and let $a \in N_L(\varphi; t^-)$. Then, $\varphi^-(a) \leq t^-$. Since $\varphi$ is a bipolar fuzzy UP-filter of $A$, we have $\varphi^-(0) \leq \varphi^-(a) \leq t^-$. Thus, $0 \in N_L(\varphi; t^-)$. Next, let $x, y \in A$ be such that $x \cdot y \in N_L(\varphi; t^-)$ and $x \in N_L(\varphi; t^-)$. Then, $\varphi^-(x \cdot y) \leq t^-$ and $\varphi^-(x) \leq t^-$. Since $\varphi$ is a bipolar fuzzy UP-filter of $A$, we have

$$\varphi^-(y) \leq \max\{\varphi^-(x \cdot y), \varphi^-(x)\} \leq t^-.$$

So, $y \in N_L(\varphi; t^-)$. Hence, $N_L(\varphi; t^-)$ is a UP-filter of $A$. Let $t^+ \in [0, 1]$ be such that $P_U(\varphi; t^+) \neq \emptyset$ and let $a \in P_U(\varphi; t^+)$. Then, $\varphi^+(a) \geq t^+$. Since $\varphi$ is a bipolar fuzzy UP-filter of $A$, we have $\varphi^+(0) \geq \varphi^+(a) \geq t^+$. Thus, $0 \in P_U(\varphi; t^+)$. Next, let $x, y \in A$ be such that $x \cdot y \in P_U(\varphi; t^+)$ and $x \in P_U(\varphi; t^+)$. Then, $\varphi^+(x \cdot y) \geq t^+$ and $\varphi^+(x) \geq t^+$. Since $\varphi$ is a bipolar fuzzy UP-filter of $A$, we have

$$\varphi^+(y) \geq \min\{\varphi^+(x \cdot y), \varphi^+(x)\} \geq t^+.$$

So, $y \in P_U(\varphi; t^+)$. Hence, $P_U(\varphi; t^+)$ is a UP-filter of $A$.

Conversely, assume that for all $t^- \in [0, 1]$, $N_L(\varphi; t^-)$ is a UP-filter of $A$ if $N_L(\varphi; t^-)$ is nonempty and for all $t^+ \in [0, 1]$, $P_U(\varphi; t^+)$ is a UP-filter of $A$ if $P_U(\varphi; t^+)$ is nonempty. Let $x \in A$. Then, $\varphi^-(x) \in [-1, 0]$. Choose $t^- = \varphi^-(x)$. Then, $\varphi^-(x) \leq t^-$, that is, $x \in N_L(\varphi; t^-) \neq \emptyset$. By assumption, we have $N_L(\varphi; t^-)$ is a UP-filter of $A$. So, $0 \in N_L(\varphi; t^-)$. Hence, $\varphi^-(0) \leq t^- = \varphi^-(x)$. Next, let $x, y \in A$. Then, $\varphi^-(x \cdot y), \varphi^-(x) \in [-1, 0]$. Choose $t^- = \max\{\varphi^-(x \cdot y), \varphi^-(x)\}$. Then, $\varphi^-(x \cdot y) \leq t^-$ and $\varphi^-(x) \leq t^-$, that is, $x \cdot y, x \in N_L(\varphi; t^-) \neq \emptyset$. By assumption, we have $N_L(\varphi; t^-)$ is a UP-filter of $A$. So, $y \in N_L(\varphi; t^-)$, Hence, $\varphi^-(y) \leq t^- = \max\{\varphi^-(x), \varphi^-(x \cdot y)\}$. Let $x \in A$. Then, $\varphi^+(x) \in [0, 1]$. Choose $t^+ = \varphi^+(x)$. Then, $\varphi^+(x) \geq t^+$, that is, $x \in P_U(\varphi; t^+) \neq \emptyset$. By assumption, we have $P_U(\varphi; t^+)$ is a UP-filter of $A$. So, $0 \in P_U(\varphi; t^+)$. Hence, $\varphi^+(0) \geq t^+ = \varphi^+(x)$. Next, let $x, y \in A$. Then, $\varphi^+(x \cdot y), \varphi^+(x) \in [0, 1]$. Choose $t^+ = \min\{\varphi^+(x \cdot y), \varphi^+(x)\}$. Then, $\varphi^+(x \cdot y) \geq t^+$ and $\varphi^+(x) \geq t^+$, that is, $x \cdot y, x \in P_U(\varphi; t^+) \neq \emptyset$. By assumption, we have $P_U(\varphi; t^+)$ is a UP-filter of $A$. So, $y \in P_U(\varphi; t^+)$. Hence, $\varphi^+(y) \geq t^+ = \min\{\varphi^+(x), \varphi^+(x \cdot y)\}$. Therefore, $\varphi$ is a bipolar fuzzy UP-filter of $A$. □

**Corollary 2.** *If $\varphi = (A; \varphi^-, \varphi^+)$ is a bipolar fuzzy UP-filter of $A$, Then, for all $k \in [0, 1]$, $C(\varphi; k)$ is a UP-filter of $A$ while $C(\varphi; k)$ is nonempty.*

**Proof.** It is straightforward by Theorems 1 and 7. □

**Theorem 8.** *Let $\varphi = (A; \varphi^-, \varphi^+)$ be a bipolar fuzzy set in $A$. Then, $\varphi = (A; \varphi^-, \varphi^+)$ is a bipolar fuzzy UP-ideal of $A$ if and only if the following statements are valid:*

*(1)   for all $t^- \in [-1, 0]$, $N_L(\varphi; t^-)$ is a UP-ideal of $A$ if $N_L(\varphi; t^-)$ is nonempty, and*
*(2)   for all $t^+ \in [0, 1]$, $P_U(\varphi; t^+)$ is a UP-ideal of $A$ if $P_U(\varphi; t^+)$ is nonempty.*

**Proof.** Assume that $\varphi$ is a bipolar fuzzy UP-ideal of $A$. Let $t^- \in [-1, 0]$ be such that $N_L(\varphi; t^-) \neq \emptyset$ and let $a \in N_L(\varphi; t^-)$. Then, $\varphi^-(a) \leq t^-$. Since $\varphi$ is a bipolar fuzzy UP-ideal of $A$, we have $\varphi^-(0) \leq \varphi^-(a) \leq t^-$. Thus, $0 \in N_L(\varphi; t^-)$. Next, let $x, y, z \in A$ be such that $x \cdot (y \cdot z) \in N_L(\varphi; t^-)$ and $y \in N_L(\varphi; t^-)$. Then, $\varphi^-(x \cdot (y \cdot z)) \leq t^-$ and $\varphi^-(y) \leq t^-$. Since $\varphi$ is a bipolar fuzzy UP-ideal of $A$, we have

$$\varphi^-(x \cdot z) \leq \max\{\varphi^-(x \cdot (y \cdot z)), \varphi^-(y)\} \leq t^-.$$

So, $x \cdot z \in N_L(\varphi; t^-)$. Hence, $N_L(\varphi; t^-)$ is a UP-ideal of $A$. Let $t^+ \in [0, 1]$ be such that $P_U(\varphi; t^+) \neq \emptyset$ and let $a \in P_U(\varphi; t^+)$. Then, $\varphi^+(a) \geq t^+$. Since $\varphi$ is a bipolar fuzzy UP-ideal of $A$, we have $\varphi^+(0) \geq \varphi^+(a) \geq t^+$. Thus, $0 \in P_U(\varphi; t^+)$. Next, let $x, y, z \in A$ be such that $x \cdot (y \cdot z) \in P_U(\varphi; t^+)$ and $y \in P_U(\varphi; t^+)$. Then, $\varphi^+(x \cdot (y \cdot z)) \geq t^+$ and $\varphi^+(y) \geq t^+$. Since $\varphi$ is a bipolar fuzzy UP-ideal of $A$, we have

$$\varphi^+(x \cdot z) \geq \min\{\varphi^+(x \cdot (y \cdot z)), \varphi^+(y)\} \geq t^+.$$

So, $x \cdot z \in P_U(\varphi; t^+)$. Hence, $P_U(\varphi; t^+)$ is a UP-ideal of $A$.

Conversely, assume that for all $t^- \in [0,1], N_L(\varphi; t^-)$ is a UP-ideal of $A$ if $N_L(\varphi; t^-)$ is nonempty and for all $t^+ \in [0,1], P_U(\varphi; t^+)$ is a UP-ideal of $A$ if $P_U(\varphi; t^+)$ is nonempty. Let $x \in A$. Then, $\varphi^-(x) \in [-1,0]$. Choose $t^- = \varphi^-(x)$. Then, $\varphi^-(x) \leq t^-$, that is, $x \in N_L(\varphi; t^-) \neq \varnothing$. By assumption, we have $N_L(\varphi; t^-)$ is a UP-ideal of $A$. So, $0 \in N_L(\varphi; t^-)$. Hence, $\varphi^-(0) \leq t^- = \varphi^-(x)$. Next, let $x, y, z \in A$. Then, $\varphi^-(x \cdot (y \cdot z)), \varphi^-(y) \in [-1,0]$. Choose $t^- = \max\{\varphi^-(x \cdot (y \cdot z)), \varphi^-(y)\}$. Then, $\varphi^-(x \cdot (y \cdot z)) \leq t^-$ and $\varphi^-(y) \leq t^-$, that is, $x \cdot (y \cdot z), y \in N_L(\varphi; t^-) \neq \varnothing$. By assumption, we have $N_L(\varphi; t^-)$ is a UP-ideal of $A$. So, $x \cdot z \in N_L(\varphi; t^-)$. Hence, $\varphi^-(x \cdot z) \leq t^- = \max\{\varphi^-(x \cdot (y \cdot z)), \varphi^-(y)\}$. Let $x \in A$. Then, $\varphi^+(x) \in [0,1]$. Choose $t^+ = \varphi^+(x)$. Then, $\varphi^+(x) \geq t^+$, that is, $x \in P_U(\varphi; t^+) \neq \varnothing$. By assumption, we have $P_U(\varphi; t^+)$ is a UP-ideal of $A$. So, $0 \in P_U(\varphi; t^+)$. Hence, $\varphi^+(0) \geq t^+ = \varphi^+(x)$. Next, let $x, y, z \in A$. Then, $\varphi^+(x \cdot (y \cdot z)), \varphi^+(y) \in [0,1]$. Choose $t^+ = \min\{\varphi^+(x \cdot (y \cdot z)), \varphi^+(y)\}$. Then, $\varphi^+(x \cdot (y \cdot z)) \geq t^+$ and $\varphi^+(y) \geq t^+$, that is, $x \cdot (y \cdot z), y \in P_U(\varphi; t^+) \neq \varnothing$. By assumption, we have $P_U(\varphi; t^+)$ is a UP-ideal of $A$. So, $x \cdot z \in P_U(\varphi; t^+)$. Hence, $\varphi^+(x \cdot z) \geq t^+ = \min\{\varphi^+(x \cdot (y \cdot z)), \varphi^+(y)\}$. Therefore, $\varphi$ is a bipolar fuzzy UP-ideal of $A$.　□

**Corollary 3.** *If $\varphi = (A; \varphi^-, \varphi^+)$ is a bipolar fuzzy UP-ideal of $A$, Then, for all $k \in [0,1], C(\varphi; k)$ is a UP-ideal of $A$ while $C(\varphi; k)$ is nonempty.*

**Proof.** It is straightforward by Theorems 1 and 8.　□

Give an example of conflict that the converse of Corollary 2, 1, and 3 is not true.

**Example 10.** *Consider a UP-algebra $A = \{0,1,2,3\}$ with the following Cayley table:*

| · | 0 | 1 | 2 | 3 |
|---|---|---|---|---|
| 0 | 0 | 1 | 2 | 3 |
| 1 | 0 | 0 | 2 | 2 |
| 2 | 0 | 1 | 0 | 2 |
| 3 | 0 | 1 | 0 | 0 |

*Define a bipolar fuzzy set $\varphi = (A; \varphi^-, \varphi^+)$ in A as follows:*

| $A$ | 0 | 1 | 2 | 3 |
|---|---|---|---|---|
| $\varphi^-$ | $-0.6$ | $-0.6$ | $-0.3$ | $-0.6$ |
| $\varphi^+$ | $0.6$ | $0.3$ | $0.6$ | $0.3$ |

*Then, for all $k \in [0,1], C(\varphi; k)$ is a UP-subalgebra (resp., UP-filter, UP-ideal) of A while $C(\varphi; k)$ is nonempty. Indeed,*

*(1)　$C(\varphi; k) = A$ if $k \in [0, 0.3]$,*
*(2)　$C(\varphi; k) = \{0\}$ if $k \in (0.3, 0.6]$, and*
*(3)　$C(\varphi; k) = \varnothing$ if $k \in (0.6, 1]$.*

*However, $\varphi$ is not a bipolar fuzzy UP-subalgebra of A. Indeed,*

$$\varphi^-(1 \cdot 3) = -0.3 > -0.6 = \max\{-0.6, -0.6\} = \max\{\varphi^-(1), \varphi^-(3)\}.$$

*By Theorems 4 and 5, we have that $\varphi$ is not a bipolar fuzzy UP-filter and a bipolar fuzzy UP-ideal of A.*

**Theorem 9.** *Let $\varphi = (A; \varphi^-, \varphi^+)$ be a bipolar fuzzy set in A. Then, $\varphi = (A; \varphi^-, \varphi^+)$ is a bipolar fuzzy strongly UP-ideal of A if and only if the following statements are valid:*

*(1)　for all $t^- \in [-1, 0], N_L(\varphi; t^-)$ is a strongly UP-ideal of A if $N_L(\varphi; t^-)$ is nonempty, and*
*(2)　for all $t^+ \in [0, 1], P_U(\varphi; t^+)$ is a strongly UP-ideal of A if $P_U(\varphi; t^+)$ is nonempty.*

**Proof.** Assume that $\varphi$ is a bipolar fuzzy strongly UP-ideal of $A$. Let $t^- \in [-1, 0]$ be such that $N_L(\varphi; t^-) \neq \varnothing$ and let $a \in N_L(\varphi; t^-)$. Then, $\varphi^-(a) \leq t^-$. Since $\varphi$ is a bipolar fuzzy strongly UP-ideal of $A$, we have $\varphi^-(0) \leq \varphi^-(a) \leq t^-$. Thus, $0 \in N_L(\varphi; t^-)$. Next, let $x, y, z \in A$ be such that $(z \cdot y) \cdot (z \cdot x) \in N_L(\varphi; t^-)$ and $y \in N_L(\varphi; t^-)$. Then, $\varphi^-((z \cdot y) \cdot (z \cdot x)) \leq t^-$ and $\varphi^-(y) \leq t^-$. Since $\varphi$ is a bipolar fuzzy strongly UP-ideal of $A$, we have

$$\varphi^-(x) \leq \max\{\varphi^-((z \cdot y) \cdot (z \cdot x)), \varphi^-(y)\} \leq t^-.$$

So, $x \in N_L(\varphi; t^-)$. Hence, $N_L(\varphi; t^-)$ is a strongly UP-ideal of $A$. Let $t^+ \in [0, 1]$ be such that $P_U(\varphi; t^+) \neq \varnothing$ and let $a \in P_U(\varphi; t^+)$. Then, $\varphi^+(a) \geq t^+$. Since $\varphi$ is a bipolar fuzzy strongly UP-ideal of $A$, we have $\varphi^+(0) \geq \varphi^+(a) \geq t^+$. Thus, $0 \in P_U(\varphi; t^+)$. Next, let $x, y, z \in A$ be such that $(z \cdot y) \cdot (z \cdot x) \in P_U(\varphi; t^+)$ and $y \in P_U(\varphi; t^+)$. Then, $\varphi^+((z \cdot y) \cdot (z \cdot x)) \geq t^+$ and $\varphi^+(y) \geq t^+$. Since $\varphi$ is a bipolar fuzzy strongly UP-ideal of $A$, we have

$$\varphi^+(x) \geq \min\{\varphi^+((z \cdot y) \cdot (z \cdot x)), \varphi^+(y)\} \geq t^+.$$

So, $x \in P_U(\varphi; t^+)$. Hence, $P_U(\varphi; t^+)$ is a strongly UP-ideal of $A$.

Conversely, assume that for all $t^- \in [0, 1]$, $N_L(\varphi; t^-)$ is a strongly UP-ideal of $A$ if $N_L(\varphi; t^-)$ is nonempty and for all $t^+ \in [0, 1]$, $P_U(\varphi; t^+)$ is a strongly UP-ideal of $A$ if $P_U(\varphi; t^+)$ is nonempty. Let $x \in A$. Then, $\varphi^-(x) \in [-1, 0]$. Choose $t^- = \varphi^-(x)$. Then, $\varphi^-(x) \leq t^-$, that is, $x \in N_L(\varphi; t^-) \neq \varnothing$. By assumption, we have $N_L(\varphi; t^-)$ is a strongly UP-ideal of $A$. So, $0 \in N_L(\varphi; t^-)$. Hence, $\varphi^-(0) \leq t^- = \varphi^-(x)$. Next, let $x, y, z \in A$. Then, $\varphi^-((z \cdot y) \cdot (z \cdot x)), \varphi^-(y) \in [-1, 0]$. Choose $t^- = \max\{\varphi^-((z \cdot y) \cdot (z \cdot x)), \varphi^-(y)\}$. Then, $\varphi^-((z \cdot y) \cdot (z \cdot x)) \leq t^-$ and $\varphi^-(y) \leq t^-$, that is, $(z \cdot y) \cdot (z \cdot x), y \in N_L(\varphi; t^-) \neq \varnothing$. By assumption, we have $N_L(\varphi; t^-)$ is a strongly UP-ideal of $A$. So, $x \in N_L(\varphi; t^-)$. Hence, $\varphi^-(x) \leq t^- = \max\{\varphi^-((z \cdot y) \cdot (z \cdot x)), \varphi^-(y)\}$. Let $x \in A$. Then, $\varphi^+(x) \in [0, 1]$. Choose $t^+ = \varphi^+(x)$. Then, $\varphi^+(x) \geq t^+$, that is, $x \in P_U(\varphi; t^+) \neq \varnothing$. By assumption, we have $P_U(\varphi; t^+)$ is a strongly UP-ideal of $A$. So, $0 \in P_U(\varphi; t^+)$. Hence, $\varphi^+(0) \geq t^+ = \varphi^+(x)$. Next, let $x, y, z \in A$. Then, $\varphi^+((z \cdot y) \cdot (z \cdot x)), \varphi^+(y) \in [0, 1]$. Choose $t^+ = \min\{\varphi^+((z \cdot y) \cdot (z \cdot x)), \varphi^+(y)\}$. Then, $\varphi^+((z \cdot y) \cdot (z \cdot x)) \geq t^+$ and $\varphi^+(y) \geq t^+$, that is, $(z \cdot y) \cdot (z \cdot x), y \in P_U(\varphi; t^+) \neq \varnothing$. By assumption, we have $P_U(\varphi; t^+)$ is a strongly UP-ideal of $A$. So, $x \in P_U(\varphi; t^+)$. Hence, $\varphi^+(x) \geq t^+ = \min\{\varphi^+((z \cdot y) \cdot (z \cdot x)), \varphi^+(y)\}$. Therefore, $\varphi$ is a bipolar fuzzy strongly UP-ideal of $A$. $\square$

**Corollary 4.** *Let $\varphi = (A; \varphi^-, \varphi^+)$ be a bipolar fuzzy set in $A$. Then, $\varphi = (A; \varphi^-, \varphi^+)$ is a bipolar fuzzy strongly UP-ideal of $A$ if and only if for all $k \in [0, 1]$, $C(\varphi; k)$ is a strongly UP-ideal of $A$ while $C(\varphi; k)$ is nonempty.*

**Proof.** It is straightforward by Theorems 1, 9, and 2, and $A$ is the only one strongly UP-ideal of itself. $\square$

**Theorem 10.** *Let $\varphi = (A; \varphi^-, \varphi^+)$ be a bipolar fuzzy UP-filter of $A$ satisfies the following assertion:*

$$x \cdot (y \cdot z) = y \cdot (x \cdot z) \text{ for all } x, y, z \in A. \tag{14}$$

*Then, $\varphi = (A; \varphi^-, \varphi^+)$ is a bipolar fuzzy UP-ideal of $A$.*

**Proof.** For all $x, y, z \in A$,

$$\varphi^-(0) \leq \varphi^-(x)$$

and

$$\begin{aligned}
\varphi^-(x \cdot z) &\leq \max\{\varphi^-(y \cdot (x \cdot z)), \varphi^-(y)\} \\
&= \max\{\varphi^-(x \cdot (y \cdot z)), \varphi^-(y)\},
\end{aligned}$$ ((14))

and

$$\varphi^+(0) \geq \varphi^+(x)$$

and

$$\begin{aligned}
\varphi^+(x \cdot z) &\geq \min\{\varphi^+(y \cdot (x \cdot z)), \varphi^+(y)\} \\
&= \min\{\varphi^+(x \cdot (y \cdot z)), \varphi^+(y)\}.
\end{aligned}$$ ((14))

Hence, $\varphi = (A; \varphi^-, \varphi^+)$ is a bipolar fuzzy UP-ideal of $A$. $\square$

**Definition 7.** *Let $\varphi = (A; \varphi^-, \varphi^+)$ be a bipolar fuzzy set in A. We define a subset $\varphi^{-1}(0,0)$ of A by*

$$\varphi^{-1}(0,0) = \{x \in A \mid \varphi^-(x) = \varphi^-(0) \text{ and } \varphi^+(x) = \varphi^+(0)\}.$$

**Theorem 11.** *Let $\varphi = (A; \varphi^-, \varphi^+)$ be a bipolar fuzzy UP-subalgebra of A. Then, $\varphi^{-1}(0,0)$ is a UP-subalgebra of A.*

**Proof.** Clearly, $0 \in \varphi^{-1}(0,0)$. Let $x, y \in \varphi^{-1}(0,0)$. Then, $\varphi^-(x) = \varphi^-(0), \varphi^+(x) = \varphi^+(0), \varphi^-(y) = \varphi^-(0)$, and $\varphi^+(y) = \varphi^+(0)$. Thus,

$$\begin{aligned}
\varphi^-(0) &\leq \varphi^-(x \cdot y) \\
&\leq \max\{\varphi^-(x), \varphi^-(y)\} \\
&= \max\{\varphi^-(0), \varphi^-(0)\} \\
&= \varphi^-(0)
\end{aligned}$$

and

$$\begin{aligned}
\varphi^+(0) &\geq \varphi^+(x \cdot y) \\
&\geq \min\{\varphi^+(x), \varphi^+(y)\} \\
&= \min\{\varphi^+(0), \varphi^+(0)\} \\
&= \varphi^+(0).
\end{aligned}$$

So, $\varphi^-(x \cdot y) = \varphi^-(0)$ and $\varphi^+(x \cdot y) = \varphi^+(0)$, that is, $x \cdot y \in \varphi^{-1}(0,0)$. Therefore, $\varphi^{-1}(0,0)$ is a UP-subalgebra of $A$. $\square$

**Theorem 12.** *Let $\varphi = (A; \varphi^-, \varphi^+)$ be a bipolar fuzzy UP-filter of A. Then, $\varphi^{-1}(0,0)$ is a UP-filter of A.*

**Proof.** Clearly, $0 \in \varphi^{-1}(0,0)$. Let $x, y \in A$ be such that $x \cdot y \in \varphi^{-1}(0,0)$ and $x \in \varphi^{-1}(0,0)$. Then, $\varphi^-(x) = \varphi^-(0), \varphi^+(x) = \varphi^+(0), \varphi^-(x \cdot y) = \varphi^-(0)$, and $\varphi^+(x \cdot y) = \varphi^+(0)$. Thus,

$$\begin{aligned}
\varphi^-(0) &\leq \varphi^-(y) \\
&\leq \max\{\varphi^-(x \cdot y), \varphi^-(x)\} \\
&= \max\{\varphi^-(0), \varphi^-(0)\} \\
&= \varphi^-(0)
\end{aligned}$$

and

$$\varphi^+(0) \geq \varphi^+(y)$$
$$\geq \min\{\varphi^+(x \cdot y), \varphi^+(x)\}$$
$$= \min\{\varphi^+(0), \varphi^+(0)\}$$
$$= \varphi^+(0).$$

So, $\varphi^-(y) = \varphi^-(0)$ and $\varphi^+(y) = \varphi^+(0)$, that is, $y \in \varphi^{-1}(0,0)$. Therefore, $\varphi^{-1}(0,0)$ is a UP-filter of $A$. $\square$

**Theorem 13.** *Let $\varphi = (A; \varphi^-, \varphi^+)$ be a bipolar fuzzy UP-ideal of A. Then, $\varphi^{-1}(0,0)$ is a UP-ideal of A.*

**Proof.** Clearly, $0 \in \varphi^{-1}(0,0)$. Let $x, y, z \in A$ be such that $x \cdot (y \cdot z) \in \varphi^{-1}(0,0)$ and $y \in \varphi^{-1}(0,0)$. Then, $\varphi^-(x \cdot (y \cdot z)) = \varphi^-(0), \varphi^+(x \cdot (y \cdot z)) = \varphi^+(0), \varphi^-(y) = \varphi^-(0)$, and $\varphi^+(y) = \varphi^+(0)$. Thus,

$$\varphi^-(0) \leq \varphi^-(x \cdot z)$$
$$\leq \max\{\varphi^-(x \cdot (y \cdot z)), \varphi^-(y)\}$$
$$= \max\{\varphi^-(0), \varphi^-(0)\}$$
$$= \varphi^-(0)$$

and

$$\varphi^+(0) \geq \varphi^+(x \cdot z)$$
$$\geq \min\{\varphi^+(x \cdot (y \cdot z)), \varphi^+(y)\}$$
$$= \min\{\varphi^+(0), \varphi^+(0)\}$$
$$= \varphi^+(0).$$

So, $\varphi^-(x \cdot z) = \varphi^-(0)$ and $\varphi^+(x \cdot z) = \varphi^+(0)$, that is, $x \cdot z \in \varphi^{-1}(0,0)$. Therefore, $\varphi^{-1}(0,0)$ is a UP-ideal of $A$. $\square$

Give an example of conflict that the converse of Theorems 11–13 is not true.

**Example 11.** *From Example 10, we have $\varphi^{-1}(0,0) = \{0\}$ is a UP-subalgebra (resp., UP-filter, UP-ideal) of A but $\varphi$ is not a bipolar fuzzy UP-subalgebra (resp., bipolar fuzzy UP-ideal, bipolar fuzzy UP-filter) of A.*

**Theorem 14.** *Let $\varphi = (A; \varphi^-, \varphi^+)$ be a bipolar fuzzy set in A. Then, $\varphi = (A; \varphi^-, \varphi^+)$ is a bipolar fuzzy strongly UP-ideal of A if and only if $\varphi^{-1}(0,0)$ is a strongly UP-ideal of A.*

**Proof.** It is straightforward by Theorem 2, and $A$ is the only one strongly UP-ideal of itself. $\square$

## 5. Conclusions and Future Work

In this paper, we have introduced the notions of bipolar fuzzy UP-subalgebras (resp., bipolar fuzzy UP-filters, bipolar fuzzy UP-ideals, and bipolar fuzzy strongly UP-ideals) of UP-algebras, proved their generalizations, and investigated some of their important properties. Then, we have the generalization diagram of bipolar-valued fuzzy sets in UP-algebras in Figure 1.

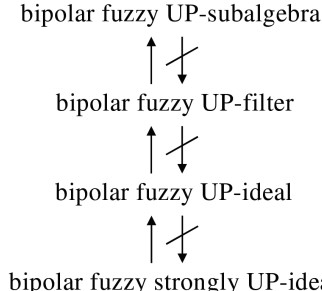

**Figure 1.** Bipolar-valued fuzzy sets in UP-algebras.

In our future study of UP-algebras, the following objectives considered:

- To get more results in bipolar-valued fuzzy sets in UP-algebras.
- To define bipolar fuzzy translations in UP-algebras.
- To define operations of bipolar-valued fuzzy sets in UP-algebras.
- To define bipolar fuzzy soft sets over UP-algebras.

**Author Contributions:** Conceptualization, A.I.; Investigation, K.K., C.U. and A.I.; Writing—original draft, K.K., C.U. and A.I.; Writing—review & editing, A.I.

**Acknowledgments:** This work was financially supported by the University of Phayao.

**Conflicts of Interest:** The authors declare no conflict of interest.

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
