# Peer review of "Bipolar Fuzzy UP-Algebras"

_mca, doi:10.3390/mca23040069_

Reviewer 1 Report

The results of this paper are correct and clear. In my opinion the paper makes a significant contribution to the study of bipolar fuzzy UP-algebras. These results may be helpful for the researchers who are working in this and other related areas. In conclusion, I recommend this article to be published in Math. Comput. Appl. But the paper needs the following corrections/changes:

  1. Introduction section of the paper should be rewritten. Authors should concentrate on the title and topic of the current paper in the introduction.

  2. The list of references should be revised and authors should be focused on the references containing of bipolar fuzzy sets, Up-algebras and also some other related references in the list.

  3. The following reference should be added in the list of references and accordingly in should be mentioned in the introduction section:

    Cubic set structure applied in UP-algebras,  Discrete Mathematics, Algorithms and Applications, 10(4) (2018), https://doi.org/10.1142/S1793830918500490.

    Representation of UP-algebras in interval-valued  intuitionistic fuzzy environment,  Italian Journal of Pure and Applied Mathematics, 38 (2017), 497-518.

    On Bipolar Fuzzy B-Subalgebras of B-Algebras, Emerging Research on Applied Fuzzy Sets and Intuitionistic Fuzzy Matrices, IGI Global, Hershey, Pennsylvania, November 2016.

    Atanassov’s intuitionistic fuzzy bi-normed KU-ideals of a KU-algebra, Journal of Intelligent & Fuzzy Systems 30, 1169-1180.

    Atanassov's Intuitionistic Fuzzy Bi-Normed KU-Subalgebras of a KU-Algebra, Missouri Journal of Mathematical Sciences 29 (1), 92-112.

Author Response

Thank you for the helpful comment.

I've edited the paper according to your comment.

Reviewer 2 Report

Nice effort. This manuscript can be accepted in MCA after careful revision.

My Comments:

This manuscript is lack of investigation. Who did introduce the concept of Bipolar fuzzy sets first? 

I suggest that read the following papers, and cite them:

[a] Bipolar fuzzy sets and relations, a computational framework for cognitive modeling and multiagent decision analysis, In Proceedings of IEEE Conference Fuzzy Information Processing Society Biannual Conference, (1994) 305-309.

[b] Bipolar fuzzy graphs, Information Sciences, 181 (2011) 5548-5564.

[c] Bipolar fuzzy K-algebras, International Journal of Fuzzy System, 10(3) (2010) 252-258.

[d] Bipolar Fuzzy Soft Lie algebras, Quasigroups and Related Systems 21 (2013) 11-18.

What is difference between bipolar-valued fuzzy sets and  bipolar fuzzy sets?

Explain Definition 6 with example. Is this definition same for IFSs?

Expand motivation of your work in Introduction. 

Give graphical representation to show family of  UP-Algebras.

Author Response

Thank you for the helpful comment.

I've edited the paper according to your comment.

Round  2

Reviewer 2 Report

The authors have revised the paper, but there are still serious issues in this version. I will not accept this paper unless paper is revised properly. 

Comments:

To make similar, change bipolar-valued fuzzy sets by bipolar fuzzy sets in the whole paper. You have used bother terms in the paper without any logical.

Papers of Akram are not cited properly.

Cite also paper of Lee. 

Read and revise it carefully. 

Author Response

Dear Professor,

Thanks for the valuable advice again. 

I have revised the paper according to your advice

If there is any point to fix, please let me know.

Best regards,

A. Iampan

Round  3

Reviewer 2 Report

I Accept this version!